# Using DNA metabarcoding and a novel canid-specific blocking oligonucleotide to investigate the composition of animal diets of raccoon dogs (*Nyctereutes procyonoides*) inhabiting the waterside area in Korea

**Cheolwoon Woo**[1], **Priyanka Kumari**[1,2], **Kyung Yeon Eo**[3]*, **Woo-Shin Lee**[4], **Junpei Kimura**[5], **Naomichi Yamamoto**[1,2]*

**1** Department of Environmental Health Sciences, Graduate School of Public Health, Seoul National University, Seoul, Republic of Korea, **2** Institute of Health and Environment, Graduate School of Public Health, Seoul National University, Seoul, Republic of Korea, **3** Department of Animal Health and Welfare, College of Healthcare and Biotechnology, Semyung University, Jecheon, Republic of Korea, **4** Department of Forest Sciences, College of Agriculture and Life Science, Seoul National University, Seoul, Republic of Korea, **5** College of Veterinary Medicine, Seoul National University, Seoul, Republic of Korea

* vetinseoul@semyung.ac.kr (KYE); nyamamoto@snu.ac.kr (NY)

**Data Availability Statement:** The raw sequencing data is available on NCBI under the BioProject number PRJNA808874.

## Abstract

The raccoon dog (*Nyctereutes procyonoides*) is known to be an opportunistic generalist who feeds on a wide variety of foods. Historically, their diet has been investigated by morphological observation of undigested remains in feces, requiring specialized knowledge such as osteology, zoology, and phytology. Here, we used DNA metabarcoding of vertebrate 12S rRNA gene and invertebrate 16S rRNA gene to investigate their fecal contents. Additionally, we developed a blocking oligonucleotide that specifically inhibits the amplification of the canid 12S rRNA gene. We confirmed that the blocking oligonucleotide selectively inhibit the amplification of raccoon dog's DNA without significantly changing the composition of the preys' DNA. We found that the main foods of raccoon dogs in our study area, the waterside of paddy fields in Korea, were fishes such as Cyprinidae and insects such as mole crickets, which makes sense given the Korean fauna and their well-known opportunistic feeding behaviors. As a method to conveniently and objectively investigate feeding habits of raccoon dogs, this study provided baseline information on DNA metabarcoding. By using DNA metabarcoding, it is expected that the diet habits and ecology of raccoon dogs will be better understood by future research.

## 1. Introduction

The raccoon dog (*Nyctereutes procyonoides*) is a medium-size canid native to East Asia including Korea [1], and was introduced to European countries in the first half of the 20th century [2]. In Korea, the raccoon dog is among one of extant carnivoran species, including the

**Funding:** This work was supported by the General Researcher Program of the National Research Foundation of Korea (2021R1F1A1060259) (NY) (https://www.nrf.re.kr/). The funders had no role in study design, data collection and analysis, decision to publish, or preparation of the manuscript.

**Competing interests:** The authors have declared that no competing interests exist.

leopard cat, Eurasian otter, yellow-throated marten, and least weasel [3]. While many of these carnivorans are endangered in Korea [3] and classified in a vulnerable category according to the Korean Red List of Threatened Species [4], the raccoon dog maintains a decent population [5] due to its high genetic diversity, adaptability, versatility in feeding habits, and reduction in the population of its predators and competitors [6–9]. Its high adaptability has also been reported in other countries. For example, in Japan, raccoon dogs are known to inhabit urban forests, such as the Imperial Palace [10], in the middle of Tokyo, a crowded megacity. In order to elucidate the origin of its high environmental adaptability, it is necessary to further investigate into the details of its ecology and food habits.

The raccoon dog is opportunistically feeding on plants, invertebrates, fishes, amphibians, reptiles, birds, and small mammals [10–17]. Through such predation and feeding, the raccoon dog is thought to directly and indirectly affect the ecosystem [18, 19], and by investigating their predation and feeding behavior, we can gain better understandings of their involvement in ecosystem functions. To investigate their predation and feeding habits, the fecal contents are often surveyed. To date, many studies have been reported on the fecal analysis of raccoon dogs in countries such as Belarus [11], Denmark [12], Finland [13, 14], Germany [15, 16], and Japan [10, 17, 20–22]. However, all of these studies rely on morphological observations of undigested remains of fruits, seeds, hair, skin, feathers, bones, and so on. Furthermore, the morphological observation is laborious and requires expertise in morphology and osteology of a broad range of organisms eaten by raccoon dogs. In addition, the morphological observation can be difficult if species or genus level identification is warranted due to the limited taxonomic resolution.

Recently, studies on fecal analysis for dietary investigations of wildlife using DNA barcoding have been reported (e.g., [23–29]). The method using DNA barcoding has the advantage that identification is objective and does not require morphological and osteological expertise. The dietary analysis using DNA barcoding is usually performed by extracting DNA from the collected fecal samples and analyzing the sequences of the extracted DNA. For carnivores, DNA markers such as vertebrate 12S rRNA gene [30] and invertebrate 16S rRNA gene [26] are targeted and sequenced. In addition, high-throughput sequencing (HTS) technologies have also been introduced for diet analyses of wildlife, such as bats in Finland [25], brown bears in Italy [26], Eurasian otters in Korea [27], and leopard cats in China [28] and Pakistan [29]. Due to its high sensitivity and ability to generate large amounts of genetic information, HTS can provide taxonomically more detailed information on dietary profiles of wildlife.

A caveat to the DNA barcoding-based method is that if the predator under investigation is a vertebrate and universal vertebrate primers are used for PCR amplification (for preparation of DNA libraries), the predator's DNA will also be amplified. This is problematic because, with the limited sequencing resource, the inclusion of DNA reads of the predator reduces the number of DNA reads of prey animals, resulting in the reduction in sensitivity in the detection of prey animals. Therefore, it is better to suppress the PCR amplification of the predator's DNA. To selectively prevent the amplification of the unwanted predator's DNA, the method using blocking oligonucleotides was invented [31]. The blocking oligonucleotide is designed to anneal at the template site between the forward and reverse primers, and it can block amplification by modification by a hydrocarbon at its 3'-end (called C3 spacer) [32, 33]. The method using blocking oligonucleotides has been widely applied to dietary surveys of wildlife, such as brown bear [26], leopard cat [28, 29], and Eurasian otter [27]. However, as far as we notice, the blocking oligonucleotide for the raccoon dog has not been reported yet.

In this study, we aimed to 1) develop the blocking oligonucleotide for the raccoon dog, and 2) investigate diet profiles of raccoon dogs in Korea using HTS-based DNA metabarcoding. For DNA metabarcoding, we sequenced vertebrate 12S rRNA gene and invertebrate 16S

rRNA gene. For sequencing vertebrate 12S rRNA gene, the developed blocking oligonucleotide was used. The use of DNA barcoding is expected to facilitate dietary research of wildlife, elucidate their roles and functions in ecosystems, and help protect them.

## 2. Materials and methods

### 2.1. Fecal samples

This study analyzed fecal samples of raccoon dogs, which were analyzed for zoonotic pathogens in our previous study [34]. Briefly, the samples were collected in a waterside area with reclaimed paddy fields in Seosan city in Chungcheongnam-do in Korea on May 21, 2017. The sampling area is close to an artificial freshwater lake named Ganwol-ho and surrounded by agricultural landscape. A total of 15 samples presumed to be raccoon dog feces were collected (S1 Fig and S1 Table in S1 File). Of them, 11 samples were confirmed to be those of raccoon dogs by the raccoon dog-specific PCR assay [35] performed in our previous study [34]. The remaining four samples were excluded from subsequent analysis. The samples were stored at −80˚C until DNA extraction. The samples were collected in a public area. Therefore, it was not necessary to obtain permission to collect samples.

### 2.2. DNA extraction

DNA was extracted from collected fecal samples by a PowerMax® Soil DNA Isolation Kit (Mobio Laboratory, Inc., Carlsbad, CA, USA). Each fecal sample was preliminary homogenized in 5 ml of ultra-pure water using a sterile wooden spatula [27]. About 0.2 g of each homogenized sample was added into a 2 ml tube of the DNA isolation kit with additional 300 mg of 0.1 mm diameter glass beads and 100 mg of 0.5 mm diameter glass beads [36]. The samples were further homogenized by a bead beater (BioSpec Products, Inc., Bartlesville, OK, USA) for 3 min. After homogenization, DNA from each fecal sample was extracted and purified by the kit's protocol and finally eluted into 50 μl of TE (10 mM Tris-HCl, 1 mM EDTA, pH = 8.0). The eluted DNA was kept at −80˚C until subsequent analysis.

### 2.3. Design of the blocking oligonucleotide RacBlk

To block unwanted amplification of the raccoon dog's DNA by universal vertebrate primers targeting 12S rRNA gene [30], we designed a canid-specific blocking oligonucleotide called RacBlk according to the method of designing blocking oligonucleotides reported by Vestheim and Jarman [31]. RacBlk has a 3-carbon spacer at its 3'-end and is designed to bind to the raccoon dog's 12S rRNA gene and block its amplification (Table 1). Specifically, RacBlk is comprised of 27 nucleotides and the first to sixth nucleotides of the RacBlk overlap with the 12SV5F, which is the forward primer for amplification of 12S rRNA gene. In addition, it was designed to avoid binding to the DNA of prey of raccoon dogs. However, since the target region of RacBlk is complementary to the sequences of other Canidae species such as *Canis lupus familiaris* (domestic dog) (Table 1), RacBlk also blocks amplification of their DNA. This was unavoidable due to the high sequence similarity of the target region of the 12S rRNA gene among species of Canidae. In Korea, four wild Canidae species have been documented: *Nyctereutes procyonoides* (raccoon dog), *Vulpes vulpes* (red fox), *Cuon alpinus* (dhole), and *Canis lupus* (Eurasian wolf) [3]. However, with the exception of the raccoon dog, these animals are highly endangered or perhaps extinct in South Korea [3]. Therefore, the chance of interaction between raccoon dogs and these canids is low. The interaction between raccoon dogs and domestic dogs may not be impossible, but due to size relationships and lifestyle differences, it is highly unlikely that raccoon dogs prey on domestic dogs.

**Table 1. Sequence of the blocking oligonucleotide RacBlk.** The 12S rRNA gene sequences of raccoon dog (*Nyctereutes procyonoides*) and its related species and potential prey were aligned with RacBlk.

| Accession number | Species (common name) | Starting position | Sequence (5′–3′) [a] | | | | | | | | | | | | | | | | | | | | | | | | | | End position |
|---|---|---|---|---|---|---|---|---|---|---|---|---|---|---|---|---|---|---|---|---|---|---|---|---|---|---|---|---|---|
| RacBlk [b] | | | C | T | C | T | A | G | A | G | G | G | A | T | A | T | A | A | A | G | C | A | C | C | G | C | C | A | A | |
| KF709435.1 | *Nyctereutes procyonoides koreensis* (raccoon dog) | 617 | · | · | · | · | · | · | · | · | · | · | · | · | · | · | · | · | · | · | · | · | · | · | · | · | · | · | · | 591 |
| NC_002008.4 | *Canis lupus familiaris* (domestic dog) | 617 | · | · | · | · | · | · | · | · | · | · | · | · | · | · | · | · | · | · | · | · | · | · | · | · | · | · | · | 591 |
| NC_008434.1 | *Vulpes vulpes* (red fox) | 617 | · | · | · | · | · | · | · | · | · | · | · | · | · | · | · | · | · | · | · | · | · | · | · | · | · | · | · | 591 |
| NC_013445.1 | *Cuon alpinus* (dhole) | 617 | · | · | · | · | · | · | C | · | · | · | · | · | · | · | · | · | · | · | · | · | · | · | · | · | · | · | · | 591 |
| NC_012374.1 | *Rattus rattus* (black rat) | 619 | · | · | · | · | · | · | · | · | A | · | · | · | · | · | · | · | · | · | A | · | · | · | · | · | · | · | · | 593 |
| NC_027932.1 | *Micromys minutus* (harvest mouse) | 614 | · | · | · | · | · | · | · | C | A | · | · | · | · | · | · | · | · | · | A | · | · | · | · | · | · | · | · | 588 |
| NC_016428.1 | *Apodemus agrarius* (striped field mouse) | 616 | · | · | · | · | · | · | · | C | A | · | · | · | · | · | · | · | · | · | A | · | · | · | · | · | · | · | · | 590 |
| NC_007236.1 | *Gallus gallus gallus* (chicken) | 1868 | · | · | · | · | · | · | · | C | A | · | · | C | · | A | · | C | C | C | · | · | · | · | · | · | · | · | · | 1842 |
| NC_006291.1 | *Carassius carassius* (crucian carp) | 1543 | · | · | · | · | · | · | C | · | · | · | · | · | · | G | · | C | · | C | T | · | · | · | · | A | · | · | C | 1517 |
| NC_015806.1 | *Silurus asotus* (Amur catfish) | 616 | · | · | · | · | · | · | · | C | A | · | · | C | · | G | · | C | · | C | T | · | · | · | · | · | · | · | · | 590 |
| KM590550.1 | *Rana coreana* (Korean brown frog) | 603 | · | · | · | · | · | · | · | A | · | · | · | C | · | C | C | C | G | T | · | G | C | A | G | T | T | · | A | 577 |
| KT878718.1 | *Pelophylax nigromaculatus* (dark-spotted frog) | 3009 | · | · | · | · | · | · | · | C | A | · | · | C | · | C | C | C | G | T | · | G | C | A | G | T | T | C | A | 2983 |

[a] The dots represent the complementary base types of the corresponding RacBlk bases.

[b] The 1st to 6th nucleotides (underlined) of RacBlk are designed to overlap the forward primer 12SV5F, which targets the 12S rRNA gene of vertebrates.

## 2.4. High-throughput DNA sequencing

For each sample, three libraries were constructed and meta-genetically analyzed. The three libraries are: (1) a library prepared with primers that target the vertebrate 12S rRNA gene with RacBlk, (2) a library prepared with primers that target the vertebrate 12S rRNA gene without RacBlk, and (3) a library prepared with primers that targets the invertebrate 16S rRNA gene. For vertebrates, 12S rRNA gene was amplified with primers 12SV5F and 12SV5R [30]. The vertebrate 12S rRNA libraries were constructed with and without the blocking oligonucleotide RacBlk. Technical duplicates were also prepared for six libraries constructed with RacBlk to test for reproducibility. For invertebrates, 16S rRNA gene was amplified with primers 16SMAV-F and 16SMAV-R [26]. Furthermore, MamMAVB1 was also included in PCR targeting the 16S rRNA gene for blocking amplification of mammal's DNA [26]. DNA sequencing was performed on an Illumina MiSeq system (Illumina, Inc., San Diego, CA, USA).

For vertebrate-specific PCR, 50 μl of each PCR reaction mixture consisted of 25 μl of Premix Taq[TM] (Takara Bio Inc., Otsu, Shiga, Japan), 0.1 μM of each of universal vertebrate primers attached to Illumina adapter sequences, 5 μM of RacBlk, 12 μl of molecular-graded water, and 1 μl of the DNA extract. PCR thermal condition was comprised of initial denaturation for 15 min at 95˚C, followed by 45 cycles of 30 s at 95˚C and 30 s at 60˚C. There was no elongation step [29]. Four concentrations of RacBlk at 2, 3, 4, and 5 μM were pretested and PCR amplification of raccoon DNA was inhibited at all of these tested concentrations (S2 Fig in S1 File). To maximize the inhibition efficiency, 5 μM was chosen as a RacBlk concentration.

For invertebrate-specific PCR, 50 μl of PCR reaction mixture consisted of 25 μl of Premix Taq™ (Takara Bio), 0.2 μM of each of universal invertebrate primers attached to Illumina adapter sequences, 2 μM of MamMAVB1, 12 μl of molecular-graded water, and 1 μl of the DNA extract. PCR thermal condition was comprised of initial denaturation for 15 min at 95˚C, followed by 55 cycles of 30 s at 95˚C and 30 s at 55˚C. There was no elongation step [26].

The resulting PCR products were purified by AMPure XP beads (Beckman Coulter, Inc., Brea, CA, USA). Next, index PCR was performed with Nextera XT Index kit (Illumina) to tag DNA libraries. The thermal cycle of index PCR comprised of 3 min at 95˚C, followed by 8 cycles of 30 s at 95˚C, 30 s at 55˚C, and 30 s at 72˚C, and final extension for 5 min at 72˚C. The tagged libraries were purified again with AMPure XP beads (Beckman Coulter). The tagged and purified libraries were quantified and normalized by Quant-iT PicoGreen dsDNA reagent kit (Life Technologies, Carlsbad, CA, USA). The normalized libraries were loaded with 30% PhiX to a v3 600 cycle-kit reagent cartridge (Illumina) for $2 \times 300$ bp paired-end sequencing on an Illumina MiSeq system.

## 2.5. Sequencing processing and analysis

The adapter and tagging sequences for MiSeq were trimmed and the reads with quality scores below 20 were removed by MiSeq Reporter version 2.5 (Illumina). Then, ambiguous bases were trimmed by Trimmomatic v 0.33 [37]. Next, OBITools [38] was used for finding unique sequences and taxonomic assignment. The *illuminapairedend* command of OBITools was used to concatenate the paired-end forward and reverse reads. The *obiuniq* command was executed to group and dereplicate the resultant reads. The cut-off for read length was set to 80 bp for the 12S rRNA gene reads and 20 bp for the 16S rRNA gene reads by the *obigrep* command. The erroneous reads were further excluded by the *obiclean* command. After quality control of sequencing reads, the resultant sequencing reads were taxonomically assigned by the *ecotag* command against the custom 12S and 16S rRNA gene reference databases. Specifically, the taxonomic annotation was performed against 12S (vertebrate) and 16S (invertebrate) rRNA genes databases generated from the latest snapshot (updated on March 13, 2022) of EMBL nucleotide sequences (http://ftp.ebi.ac.uk/pub/databases/ena/sequence/snapshot_latest/std/).

## 2.6. Statistical analysis

The statistical analysis was performed to evaluate the blocking efficacy of RacBlk on R version 4.1.0 with the phyloseq package [39] and vegan package [40]. First, we compared the number of sequence reads identified as the family Canidae (to which the raccoon dog belong) with and without RacBlk for each sample. Second, we evaluated about unintended inhibition by RacBlk. The use of blocking oligonucleotide may inhibit amplification of DNA of prey animals, resulting in change in dietary proportion measured. Therefore, we evaluated the change in dietary proportion due to the use of RacBlk. Specifically, the differences in composition of unique sequences from prey animals were compared with and without RacBlk. To analyze the composition of prey animals, the reads that were identified as Canidae were excluded. The remaining reads were rarefied into 120 reads per library, which was the smallest number of sequence reads from prey animals (excluding reads from Canidae) found in the library prepared without RacBlk. Rarefaction was necessary because the number of sequence reads from prey animals (excluding reads from Canidae) was much smaller in libraries prepared without RacBlk than those with RacBlk, and the comparison had to be done under the same condition, i.e., the same number of sequences. Note that rarefaction was performed only for the purpose to compare the sequencing results with and without RacBlk. No rarefaction was performed in other analyses. Based on the rarefied libraries, the difference in composition of unique sequences

from prey animals were analyzed. The differences were characterized based on the Bray–Curtis dissimilarity (community structure) and Jaccard index (community membership). To compare the difference in composition of unique sequences from prey animals detected with and without RacBlk, the adonis2 function in vegan package was used for performing permutational multivariate analysis of variance (PERMANOVA). The intra-sample and inter-sample variances of composition of unique sequences from prey animals were compared based on the beta dispersion calculated by the betadisper function in vegan package. The intra-sample variance was defined as the variance of the composition measured with and without RacBlk for the same sample, while the inter-sample variance was defined as the variance of composition between samples measured with (or without) RacBlk. Kruskal-Wallis rank sum tests and *post hoc* Wilcoxon rank-sum tests were used to compare the differences between the intra- and inter-sample variances.

## 3. Results

### 3.1. Sequencing statistics

From a total of 11 raccoon dog's fecal samples, 11 pairs of vertebrate libraries with and without RacBlk were obtained. Additionally, six vertebrate libraries prepared with RacBlk were technically duplicated. For invertebrates, one sample was not PCR amplified, resulting in a total of 10 libraries (S2 Table in S1 File). From 38 libraries including duplicates, a total of 4,676,053 high-quality sequence reads were obtained, consisting of 1,367,071 reads of vertebrates without RacBlk, 2,513,355 reads of vertebrates with RacBlk, and 795,627 reads of invertebrates.

### 3.2. Performance of the blocking oligonucleotide RacBlk

Fig 1A shows relative abundance of vertebrates identified at the family level. Without RacBlk, the median relative abundance of Canidae was 99.0%, ranging from 51.6% to 99.9%, suggesting that most of the sequence reads were of DNA from the host animal (i.e., raccoon dog). With RacBlk, the median relative abundance of Canidae read was 55.8%, ranging from 5.3% to 72.9%, indicating that RacBlk inhibited amplification of DNA of the host animal. The reduction was statistically significant ($p < 0.001$; Wilcoxon rank-sum tests; Fig 1B). It was also confirmed that there was no significant difference in the community structure of prey animals detected with and without RacBlk ($p > 0.05$; PERMANOVA; Fig 1C), and that the variance of the prey structure measured with and without RacBlk for the same sample (intra-sample variance) was significantly smaller than the variance of the structure between samples measured with (or without) RacBlk (inter-sample variance) ($p < 0.0001$; Wilcoxon rank-sum test; Fig 1D). The similar results were obtained for the community membership (Fig 1E and 1F). These suggest that the addition of RacBlk did not significantly change the composition of the detected prey animals. In addition, we compared the relative abundance of specific prey genera measured with and without RacBlk (Fig 1G). The differences measured with and without RacBlk were within an order of magnitude for all genera (Fig 1H). The result of reproducibility of sequencing with RacBlk based on technical duplicates is shown in S3 Fig in S1 File. The result shows high reproducibility of sequencing since the intra-sample variability was significantly smaller than the inter-sample variability.

### 3.3. Vertebrate composition

Fig 2A shows mean relative abundance of vertebrates at the family level as measured by 12S rRNA gene sequencing. The majority of vertebrate diets that were eaten by raccoon dogs in our sampling site was found to be freshwater fishes, with the families such as Cyprinidae (carp

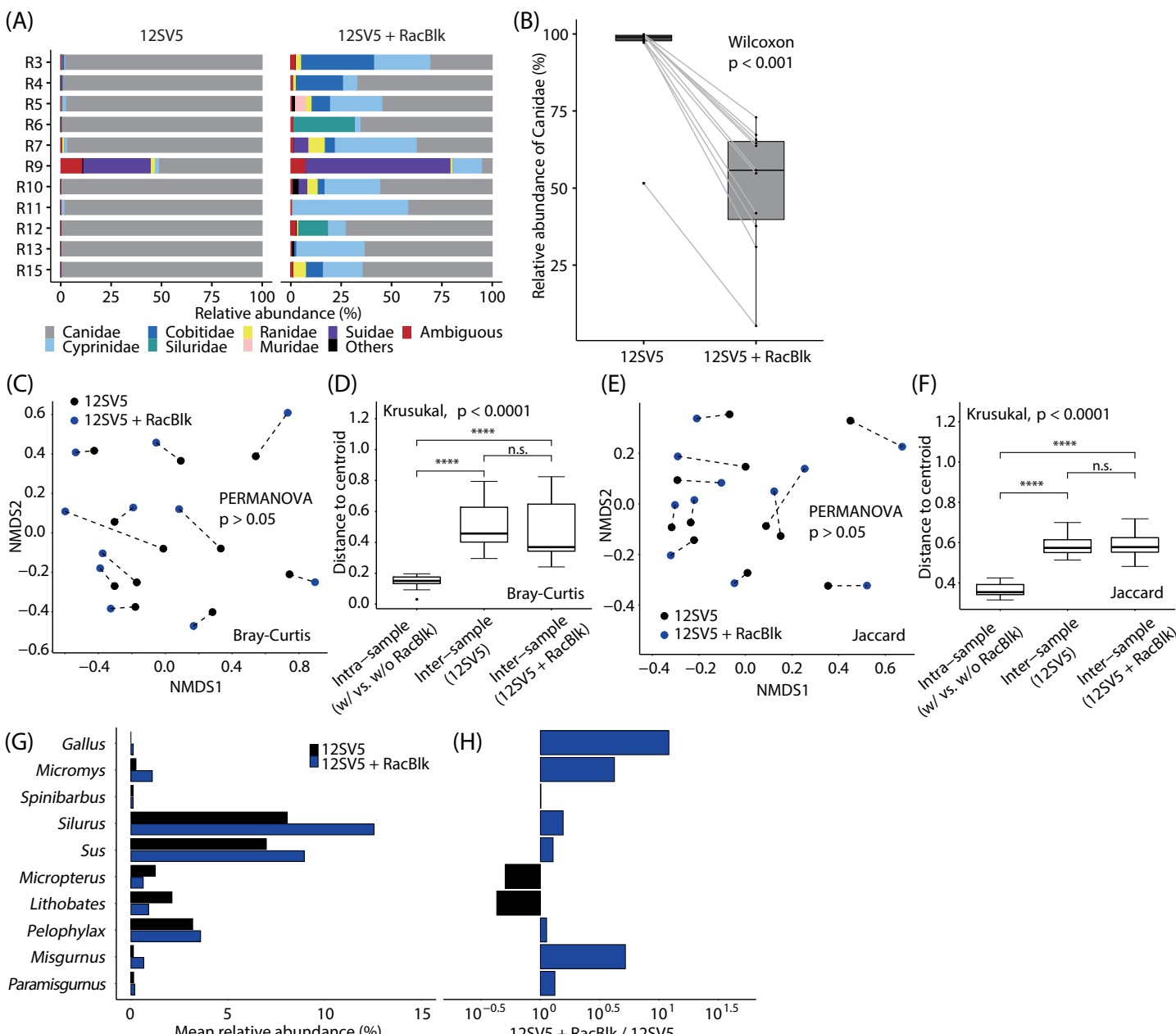

**Fig 1. Performance of blocking oligonucleotide RacBlk.** The 12S rRNA gene libraries prepared with the universal vertebrate primer pair 12SV5 are compared with and without RacBlk. (A) Relative abundance of vertebrates identified at the family level. (B) Relative abundance of Canidae to which the raccoon dog belongs. (C) Non-metric multidimensional scaling (NMDS) plot showing the Bray–Curtis dissimilarity of community structure of prey animals with and without RacBlk. (D) Boxplot showing the intra- and inter-sample variances of the Bray–Curtis dissimilarity of prey composition. (E) NMDS plot showing the Jaccard index of community membership of prey animals with and without RacBlk. (F) Boxplot showing the intra- and inter-sample variances of the Jaccard index of prey membership. (G) Mean relative abundances of the top 10 vertebrate genera detected with and without RacBlk. The reads identified as sequences of Canidae were excluded from the calculation of relative abundance. (H) Ratio of mean relative abundances measured with and without RacBlk. In the panels (C) and (E), the data from the same sample were connected by a line. In the panels (D) and (F), the four asterisks (****) represent $p < 0.0001$ by the *post hoc* Wilcoxon rank-sum test. The abbreviation "n.s." represents that there is no significant difference.

or minnows) (50.2%), Cobitidae (true loaches) (17.0%), and Siluridae (catfishes) (13.0%). Additionally, the families Ranidae (true frogs) (5.5%), Suidae (pigs or boars) (9.0%), and Muridae (rodents) (1.2%) were identified. Although most of the sequence reads identified as Cyprinidae and Cobitidae were ambiguous at the genus level, *Spinibarbus* (cyprinid fishes),

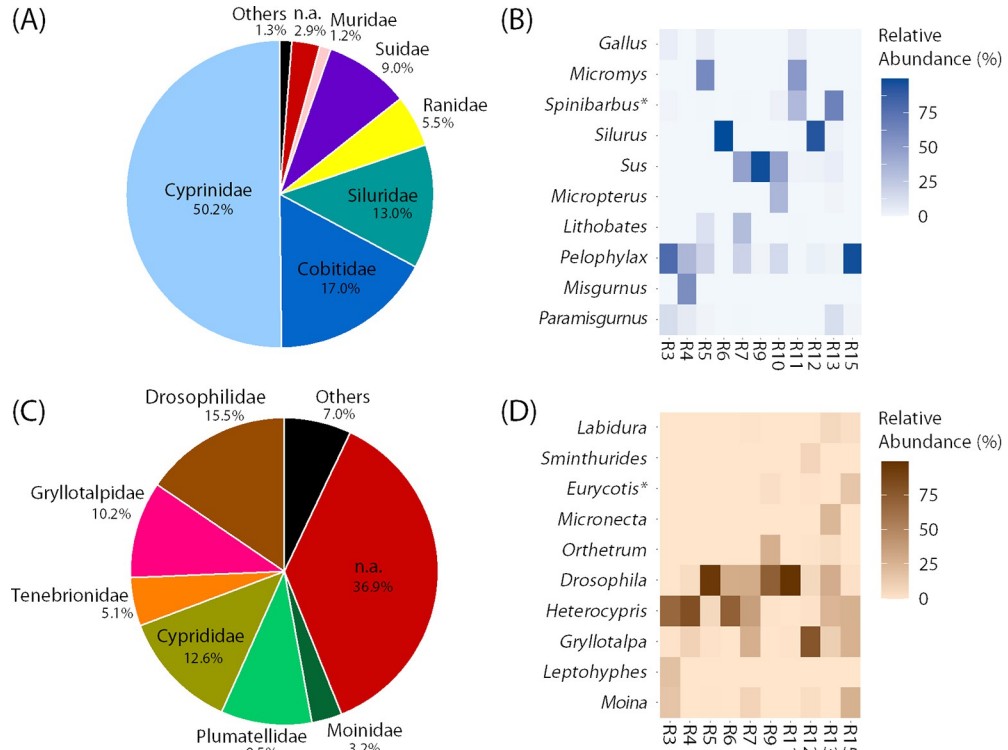

**Fig 2. Dietary composition of raccoon dogs.** (A) Mean relative abundance of vertebrates at the family level as measured by 12S rRNA gene sequencing. The data obtained using the blocking oligonucleotide RacBlk are shown. The sequence reads identified as the family Canidae were excluded from the calculation for relative abundance. (B) Relative abundance of the 10 most abundant vertebrate genera detected by 12S rRNA gene sequencing with RacBlk. The unidentified reads at the genus level and genera belong to the family Canidae were excluded from the calculation for relative abundance. (C) Mean relative abundance of invertebrates at the family level as measured by 16S rRNA gene sequencing. The sequence reads assigned to non-targeted organisms such as bacteria and vertebrates were excluded from the calculation for relative abundance. (D) Relative abundance of the 10 most abundant invertebrate genera detected by 16S rRNA gene sequencing. The unidentified reads at the genus level and reads assigned to non-targeted organisms such as bacteria and vertebrates were excluded from the calculation for relative abundance. The abbreviation "n.a." represents the ambiguous reads were not classified into specific family. The detection of *Spinibarbus* and *Eurycotis* (with *) may be misidentification (see text for details).

*Misgurnus* (true loaches), and *Paramisgurnus* (large-scale loaches) were identified (Fig 2B). Additionally, *Silurus* (catfishes) and *Micropterus* (black bass) were identified from other fish families. Most of the sequence reads of Ranidae were classified into *Pelophylax* (green frogs) or *Lithobates* (true frogs). The mammalian genera such as *Sus* (domestic pig or wild boar) and *Micromys* (harvest mouse) were also identified.

## 3.4. Invertebrate composition

Fig 2C shows mean relative abundance of invertebrates at the family level as measured by 16S rRNA gene sequencing. More than half of the sequence reads were identified at the family level. Some of the detected families were aquatic, such as Cyprididae (freshwater ostracods) (12.6%), Plumatellidae (freshwater bryozoans) (9.5%), and Moinidae (water fleas) (3.2%). Others included terrestrial organisms such as Drosophilidae (flies) (15.5%), Gryllotalpidae (mole crickets) (10.2%) and Tenebrionidae (darkling beetles) (5.1%). The sequence reads identified as Tenebrionidae and Plumatellidae were ambiguous at the genus level. However, the sequence reads identified as other families were identified down to the genus level (Fig 2D). For

instance, aquatic genera such as *Heterocypris* (freshwater ostracods) and *Moina* (water fleas) were identified. The genera of insects such as *Gryllotalpa* (mole crickets), *Orthetrum* (dragonflies), *Micronecta* (water boatmen), *Eurycotis* (Florida woods cockroach), and *Labidura* (earwigs) were also identified. *Drosophila* (flies) was also abundantly detected in many samples.

## 4. Discussion

In this study, we used DNA metabarcoding of the invertebrate 16S rRNA gene and vertebrate 12S rRNA gene with the newly developed blocking oligonucleotide RacBlk to study dietary composition of raccoon dogs inhabiting a coastal area with paddy fields in Korea in late spring. We found that vertebrate diets of raccoon dogs mainly consisted of aquatic species such as freshwater fishes, frogs, and aquatic arthropods and fossorial and terrestrial insects in our study area. The results seem reasonable given the geographical characteristics of the study area. In addition, the results seem to be consistent with previous studies showing that raccoon dogs are generalists who eat a wide range of diets available in their habitat [11, 12, 15–17, 41].

### 4.1. Fishes are dominant prey for raccoon dogs inhabiting waterside areas

The organisms identified from fecal samples of raccoon dogs in this study were in congruence with the fauna reported in South Korea. For example, vertebrate 12S rRNA gene sequencing identified freshwater fishes, such as *Silurus*, *Micropterus*, *Misgurnus*, and *Paramisgurnus*, and frogs, such as *Pelophylax* and *Lithobates*, at the genus level (Fig 2B). Notably, fishes represented 80.2% of vertebrate DNA at the family level if combining Cyprinidae (50.2%), Cobitidae (17.0%), and Siluridae (13.0%). In Korea, two species of *Silurus*, i.e., *Silurus asotus* (Amur catfish) and *Silurus microdorsalis* (slender catfish), are known to inhabit rivers and lakes [42–45]. Similarly, *Misgurnus* species, such as *Misgurnus anguillicaudatus* (pond loach) and *Misgurnus mizolepis* (mud loach), and *Paramisgurnus* species, such as *Paramisgurnus dabryanus*, a senior synonym of *M. mizolepis*, are common in Korea [46]. Additionally, *Micropterus salmoides* (largemouth bass) is known to be widely distributed in freshwater systems of South Korea [43, 45, 47]. Previous studies reported [11–16, 20] that fishes are regarded as one of important food sources for raccoon dogs especially in waterside areas (i.e., coast and/or lake areas) although these studies did not identify fish taxa due to morphological observations. Our result obtained in the coastal area in Korea is in line with the tendency reported by the existing studies conducted in other countries. Species of frogs such as *Pelophylax nigromaculatus* (dark-spotted frog) *and Lithobates catesbeianus* (= *Rana catesbeiana*) (American bullfrog) are also commonly observed in Korea [48, 49]. For instance, the dark-spotted frog inhabits freshwater such as ponds around rivers, lakes, swamps and rice paddies [50]. Studies in Finland [13, 14] and Germany [16] reported *Rana* spp. as prey for raccoon dogs. Moreover, it has been recently reported that raccoon dogs are hunting the American bullfrog [51].

Mammals such as *Sus* (wild boar) and *Micromys* (harvest mouse) were also detected by vertebrate 12S rRNA gene sequencing (Fig 2B). These mammalian species are known to inhabit South Korea [52–54]. For instance, the population of wild boar (*Sus scrofa*) is known to be rapidly increased from the 2010s even threatening urban livings in Korea [52, 53]. Although the raccoon dog is thought to be unable to directly hunt animals larger than themselves, such as wild boars and deer, they have been reported to consume their carcasses. [11, 15, 16]. The scavenging behavior of raccoon dogs is known to be especially noticeable in winter [11]. Meanwhile, this study was conducted in late spring. The smaller proportion of the family Suidae (9.0%) than those of fish families Cyprinidae (50.2%), Cobitidae (17.0%), and Siluridae (13.0%) (Fig 2A) may reflect this seasonal trend. The family Muridae represents 1.2% of

vertebrate DNA (Fig 2A), and *Micromys* was identified at the genus level (Fig 2B). Some of murid species are reported as dietary items of raccoon dogs in Denmark [12] and Finland [13].

Invertebrate 16S rRNA gene sequencing identified *Gryllotalpa* as one of the most abundant invertebrate genera (Fig 2D). In Korea, *Gryllotalpa orientalis* (mole cricket) is known to inhabit [55]. The mole cricket is known to inhabit wetlands and paddy fields by burrowing [56], and was reported to be predated by raccoon dogs by morphological observation of fecal contents in a Japanese study [22]. Furthermore, Tenebrionidae that belong to the order Coleoptera (beetles) was identified at the family level (Fig 2C). In line with the result of previous studies, the order Coleoptera were reported as one of main prey for raccoon dogs [11, 15, 16, 22, 57] and inhabited in diverse environments [58, 59]. Tenebrionidae is regarded as one of diverse groups in the order Coleoptera. In Korea, 129 species of Coleoptera are known to inhabit [60–62]. We also identified *Orthetrum*, *Micronecta*, and *Labidura* (Fig 2D). Several species belonging to those genera were reported to inhabit Korea [63–65]; however, knowledge about whether raccoon dogs feed on them is limited. Additionally, 16S rRNA gene sequencing detected small-size invertebrates such as *Heterocypris* and *Moina*. Due to their small size, these organisms are unlikely to be directly preyed on by raccoon dogs. Perhaps it was due to secondary predation, and they are thought to be accidentally swallowed or attached to larger aquatic organisms that were preyed on by raccoon dogs.

Overall, we found that the most dominant food for raccoon dogs is fishes at our sampling site located in the coastal area of Korea, representing 80.2% of vertebrate DNA. The family Ranidae was also abundant (5.5%) (Fig 2A). These results indicate that readily available freshwater fishes and frogs, rather than small mammals such as mice, are preferentially consumed by raccoon dogs in our study area and season. This seems to be in line with existing knowledge. For example, previous studies reported that raccoon dogs eat other animals, such as birds and amphibians, when small mammals are not readily available [13, 15]. Among the fishes detected, the family Cyprinidae was the most abundant and accounted for 50.2% of vertebrate DNA (Fig 2A), suggesting that they are main prey for raccoon dogs in our survey area. In Korea, Cyprinidae species, such as *Cyprinus carpio* (common carp) and *Carassius auratus* (crucian carp), abundantly exist [43–45]. We speculate that the detection of *Spinibarbus* at the genus level (Fig 2B) may be misidentification of *Cyprinus* and *Carassius* because *Spinibarbus* is known to inhabit East Asia [66] but has not been reported in South Korea. Indeed, the sequence of target region of *Spinibarbus* is identical or similar to the sequences of other Cyprinidae species (S4 Fig in S1 File), indicating the possibility of misidentification between species of Cyprinidae. Similarly, we identified *Eurycotis* by invertebrate 16S rRNA gene sequencing (Fig 2D). However, there have been no reports that *Eurycotis* inhabit Korea. Therefore, we surmise that sequence similarity caused a misidentification between *Eurycotis* and the indigenous cockroach *Periplaneta* (S5 Fig in S1 File).

## 4.2. Blocking oligonucleotide RacBlk reduces PCR amplification of the raccoon dog's DNA

In wildlife dietary surveys using PCR, not only preys' DNA but also the predator's DNA can be possibly amplified, which is problematic because it can reduce the detection sensitivity of the preys' DNA. To alleviate this problem, the idea of using the blocking oligonucleotide, which selectively inhibits PCR amplification of a predator's DNA, was invented [31]. Blocking oligonucleotides have been widely used in dietary studies of wildlife such as the leopard cat [29], Eurasian otter [27], Antarctic krill [31], brown bear [26], penguin [67], and seal [68]. In this study, we designed the blocking oligonucleotide RacBlk for the raccoon dog. RacBlk has

successfully reduced the amplification of the canid DNA in PCR that targeted the vertebrate 12S rRNA gene (Fig 1A and 1B). Furthermore, we could confirm that there was no significant change in dietary composition with and without RacBlk (Fig 1C and 1E), indicating that unintended inhibition of the amplification of preys' DNA was insignificant.

The blocking oligonucleotide RacBlk significantly reduced PCR amplification of DNA of Canidae, but the inhibition was not complete. In fact, 5.3%–72.9% sequence reads were those of Canidae even with RacBlk (Fig 1A and 1B). The reduction seems to be lower than the reduction with blocking oligonucleotides reported by previous studies. For example, blocking oligonucleotides for the leopard cat and Eurasian otter developed by previous studies reported reducing the predators' DNA down to 0% and 0.21%, respectively [27, 29]. The lower reduction efficiency may be due to the shorter length of RacBlk (27 nt) than other blocking oligonucleotides (40–50 nt). In fact, the fact that longer blocking oligonucleotides are associated with higher blocking efficiencies has been reported by previous studies [69, 70]. We do not know the mechanism behind, but one possible explanation is that a short-length blocking oligonucleotide allows annealing and extension of the universal primer to the target region of the blocking oligonucleotide before the blocking oligonucleotide is annealed to it, resulting in amplification of the target organism (which should be blocked) by the universal primers. The short length of RacBlk was unavoidable due to the limited 12S rRNA gene site with sequence specific to the raccoon dog. In addition, due to the limited targeting site, the selection of the region whose sequence is identical or similar to those of other species of Canidae was unavoidable (Table 1). However, we don't expect this to be a problem. As mentioned above, three other species of Canidae have been reported in Korea, i.e., *Vulpes vulpes*, *Cuon alpinus*, and *Canis lupus* (Eurasian wolf) [3]. However, it is highly unlikely that they inhabit the study area because they are highly endangered or extinct in South Korea [3]. We also anticipate that the interaction between raccoon dogs and domestic dogs (*Canis lupus familiaris*) is unlikely because of difference in lifestyle. However, it may not be impossible. Future research needs to elucidate their relationships.

## 4.3. Limitations of DNA metabarcoding in dietary surveys for omnivore animals

DNA barcoding is a useful tool for wildlife dietary research. However, it is not without problems, as pointed out elsewhere (e.g., [27, 71]). We recognize that this study also has such limitations. In particular, we would like to acknowledge that there is ongoing argument over whether DNA-based methods can accurately quantify the proportion of dietary content [72–74]. This study reported relative abundance rather than detected or undetected since the information of relative abundance also includes the information of detected or undetected and may provide some insights such as detection specificity and sensitivity (or insensitivity) of the sequencing method used. In addition, there may be limitations specific to dietary analysis of omnivorous animals such as raccoon dogs. First, the raccoon dog is an omnivorous carnivore that also eats plants [11, 12, 15–17, 41]. However, this study focused on the animal diets and does not report the plant diets. In our preliminary analysis, large amounts of Pinaceae DNA (~80% of total reads) was detected in fecal samples (S6 Fig in S1 File). We suspect that the detection of Pinaceae plants is due to the contamination by pollen since it is known that massive amounts of pine pollen are dispersed in spring in Korea [75]. Similarly, the detection of large amount of *Drosophila* might be due to their infestation on feces after defecation. Further validation is required for these issues of post-defecation contamination. Second, DNA metabarcoding does not allow inter-phylum comparison of abundance of vertebrates and invertebrates. Similarly, it does not allow inter-kingdom comparison of abundance of animals and

plants. The comparison has to be limited to within the phylum or kingdom. For the comparison on the same scale between phyla or between kingdoms, methods such as metagenomic sequencing are required. Lastly, there may be the lack of taxonomic resolution of the selected DNA marker, primer pair, and/or reference database depending on the detected organisms. In our study, large amounts of fishes belonging to Cyprinidae and Cobitidae were detected (Fig 2A); however, many of them were ambiguous at the genus level. In this study, we used the primer pair 12SV5F and 12SV5R targeting 12S rRNA gene of vertebrates, which is widely used to identify common vertebrate species [30, 76]. For omnivorous opportunists such as the raccoon dog, it is difficult to predict the type of foods in advance, making it difficult to preselect the most appropriate sequencing strategy (e.g., target gene and primer pair). One way to alleviate this problem is to target multiple DNA markers and/or use multiple primer pairs for a single group of organisms. However, this should be decided according to the taxonomic resolution required and the sequencing resources available.

## 5. Conclusion

Using DNA metabarcoding and a canid-specific blocking oligonucleotide developed in this study, we identified that the main foods of raccoon dogs inhabiting the waterside of paddy fields in Korea were fishes such as Cyprinidae and insects such as mole crickets. The results seem reasonable in light of the Korean fauna and their well-known opportunistic feeding behaviors. The raccoon dog, which is relatively abundant in Korea, is known not only to play a role in the ecosystem, but also to be a reservoir of zoonotic pathogens [77]. Therefore, understanding their ecology is essential not only for conservation biology but also for public health. Understanding their feeding habits helps to understand their ecology. As a method to investigate their feeding habits, this study presented the baseline information on DNA metabarcoding, which does not require specialized knowledge such as osteology. By using convenient and objective DNA barcoding, the dietary habits and ecology of raccoon dogs are expected to be better understood by future research.

## Supporting information

**S1 File. Supporting information.**
(PDF)

## Author Contributions

**Conceptualization:** Kyung Yeon Eo, Woo-Shin Lee, Junpei Kimura, Naomichi Yamamoto.

**Data curation:** Cheolwoon Woo.

**Formal analysis:** Cheolwoon Woo.

**Funding acquisition:** Naomichi Yamamoto.

**Investigation:** Cheolwoon Woo, Priyanka Kumari, Kyung Yeon Eo, Junpei Kimura, Naomichi Yamamoto.

**Project administration:** Naomichi Yamamoto.

**Resources:** Kyung Yeon Eo, Woo-Shin Lee, Junpei Kimura.

**Supervision:** Kyung Yeon Eo, Woo-Shin Lee, Junpei Kimura.

**Validation:** Naomichi Yamamoto.

**Visualization:** Cheolwoon Woo, Naomichi Yamamoto.

**Writing – original draft:** Cheolwoon Woo, Naomichi Yamamoto.

**Writing – review & editing:** Priyanka Kumari, Kyung Yeon Eo, Woo-Shin Lee, Junpei Kimura, Naomichi Yamamoto.

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
