## [Decision Letter · Decision Letter 0]

7 Apr 2022

PONE-D-22-06464Using DNA metabarcoding to investigate the composition of animal diets of raccoon dogs (Nyctereutes procyonoides) inhabiting the waterside area in KoreaPLOS ONE

Dear Dr. Yamamoto,

Thank you for submitting your manuscript to PLOS ONE. After careful consideration, we feel that it has merit but does not fully meet PLOS ONE’s publication criteria as it currently stands. Therefore, we invite you to submit a revised version of the manuscript that addresses the points raised during the review process.

We look forward to receiving your revised manuscript.

Kind regards,

Bi-Song Yue, Ph.D

Academic Editor

PLOS ONE

Journal Requirements:

[This work was supported by the General Researcher Program of the National Research Foundation of Korea (2021R1F1A1060259) (NY).]

 [This work was supported by the General Researcher Program of the National Research Foundation of Korea (2021R1F1A1060259) (NY) (https://www.nrf.re.kr/). The funders had no role in study design, data collection and analysis, decision to publish, or preparation of the manuscript.]

a) You may seek permission from the original copyright holder of Figure 1 to publish the content specifically under the CC BY 4.0 license.  

Reviewers' comments:

Reviewer's Responses to Questions

**Comments to the Author**

1. Is the manuscript technically sound, and do the data support the conclusions?

Reviewer #1: Yes

Reviewer #2: Partly

2. Has the statistical analysis been performed appropriately and rigorously? 

Reviewer #1: Yes

Reviewer #2: No

3. Have the authors made all data underlying the findings in their manuscript fully available?

Reviewer #1: Yes

Reviewer #2: Yes

4. Is the manuscript presented in an intelligible fashion and written in standard English?

Reviewer #1: Yes

Reviewer #2: Yes

5. Review Comments to the Author

Reviewer #1: The manuscript provides data on the dietary habits of raccoon dogs in Korea using HTS-based DNA metabarcoding. Additionaly presents interesting data with and without using newly developed blocking oligonucleotide RacBlk to block unnwanted amplification of the raccoon dog's DNA.

The autors presented material and methods as wellas results in a clear and very precised way. The section Statistical analysis explains what statistical test has been used and why, additionaly authors prerformed PERMANOVA,Ktuskal-wallis rank sum test and post hoc Wilcoxon rank sum test . Authors also provided raw sequencing data -available in NCBI under the project number.

The manuscript is written in standard english, no additional corrections are needed.

Summarizng, the manuscript provides valiuable information on racoon dog's died by using advanced molecular tehniques, and thus more reliable than morphological and osteological expertise. Inventing RacBlock blocking oligonucleotide for 12S rRNA gene significantly increased the value of the paper.

Comment for the authors:

Once the latin names have been given in the Results section there is no need to repaeat them in the discussion, there are somme repetitions eg.line 266: Suidae (pigs or boars) is repeated in line 348, in this case the bracket should be dropped.

Reviewer #2: This paper investigates the diet of the raccoon dog using only a very limited number of samples, but the authors have successfully generated a blocking primer which is another achievement of this work. Perhaps the title could be modified so this is more explicit.

Line 125- Please state clearly if these other canids coexist with raccoon dogs at sampled localities. This is mentioned in discussion but I think it is important for readers have this clarified earlier in the text

Line 144-146: this information is repeated and more complete in following sections

Line 150- How was the concentration of blocking primer optimized? PCR conditions are incomplete (extension?). Did the blocking primer anneal at the same temperature as the 12S universal primers? What are the sizes of the 12S and 16S libraries?

Line 156- again, PCR conditions look incomplete.

Line 168- Please state whether PhiX was used in the run

Line 176- I am not that familiar with this software but it seems minimum size of sequences considered was 80 bp for 12 S and 20 bp for 16S. I can imagine taxonomic results could be meaningful with 80 bp for 12S, but 20 bp for 16S looks impossible to me. I am also concerned about overestimation of OTUs as perhaps every unique 20bp sequence was considered as an OTU even if these 20 bp were included in larger sequences? How does the obiuniq command deal with size?

Line 183- I understand these data have some years now, but I think the authors should have used the currently available data at EMBL

Line 189- I think the comparison of libraries constructed with or without blocking primer should use data at OTU level. Otherwise important taxa or taxa of interest in future studies could be excluded and no results regarding this are reported (not available in supplementary materials). The fact that taxonomic assignment was achieved for most OUT at family level does not preclude analysis at genus and OTU level. Without this the Discussion section could be misleading as the authors clearly state: “indicating the absence of unintended inhibition of the amplification of preys’ DNA. “. I don’t understand how this was tested if analysis were performed at family level. Also why did you rarify these data? Was it to eliminate rare prey? I think metabarcoding data applied to diets, even using universal primers, implies data needs to be treated as presence/absence rather than abundance. There is a lot that cannot be controlled in a PCR reaction, namely different levels of primer affinity to each prey DNA, different digestibility of prey which can also bias DNA “quantity” etc

Line 200- Differences between libraries were evaluated using BC distance. However this accounts for abundance and what I have mentioned before I don’t think this metric should be used as it likely results in spurious results. Unless the nº of OTUs from a particular family is used as an abundance metric? There are other distances that could be applied to presence absence data (Jaccard for example).

Line239- These are the kind of biases you would expect and that is why in my view, using abundance data in this kind of analysis is not correct

Line 406- this sentence erroneously leads the reader to think these blocking primers only block racoon dog DNA. I think this should be re-phrased

Line 413- I am very interested by this idea. Can you explain further why sequence length of blocking primer should affect the level of DNA blocked? If the primer was longer it may be more specific but I do not comprehend why there should be an implication with the amount of DNA blocked.

6. PLOS authors have the option to publish the peer review history of their article (what does this mean?). If published, this will include your full peer review and any attached files.

Reviewer #1: No

Reviewer #2: No

---

## [Author Response · Author response to Decision Letter 0]

3 Jun 2022

Please find our responses to the reviewers’ comments below. The parts changed are marked in the revised manuscript, and the line numbers in our responses refer to the revised manuscript with track changes.

Editor:

RESPONSE: Thank you for your reminders. We believe the manuscript complies with PLOS ONE's style requirements. 

[This work was supported by the General Researcher Program of the National Research Foundation of Korea (2021R1F1A1060259) (NY).]

We note that you have provided funding information that is currently declared in your Funding Statement. However, funding information should not appear in the Acknowledgments section or other areas of your manuscript. We will only publish funding information present in the Funding Statement section of the online submission form. Please remove any funding-related text from the manuscript and let us know how you would like to update your Funding Statement. Currently, your Funding Statement reads as follows: [This work was supported by the General Researcher Program of the National Research Foundation of Korea (2021R1F1A1060259) (NY) (https://www.nrf.re.kr/). The funders had no role in study design, data collection and analysis, decision to publish, or preparation of the manuscript.] Please include your amended statements within your cover letter; we will change the online submission form on your behalf.

RESPONSE: The acknowledgements section was excluded.

3. We note that Figure 1 in your submission contain [map/satellite] images which may be copyrighted. All PLOS content is published under the Creative Commons Attribution License (CC BY 4.0), which means that the manuscript, images, and Supporting Information files will be freely available online, and any third party is permitted to access, download, copy, distribute, and use these materials in any way, even commercially, with proper attribution. For these reasons, we cannot publish previously copyrighted maps or satellite images created using proprietary data, such as Google software (Google Maps, Street View, and Earth). For more information, see our copyright guidelines: http://journals.plos.org/plosone/s/licenses-and-copyright. We require you to either (1) present written permission from the copyright holder to publish these figures specifically under the CC BY 4.0 license, or (2) remove the figures from your submission:

RESPONSE: The original Fig 1 including the map was removed.

RESPONSE: The information was added (Lines 702-703). 

Reviewer #1:

1. The manuscript provides data on the dietary habits of raccoon dogs in Korea using HTS-based DNA metabarcoding. Additionaly presents interesting data with and without using newly developed blocking oligonucleotide RacBlk to block unnwanted amplification of the raccoon dog's DNA. The autors presented material and methods as wellas results in a clear and very precised way. The section Statistical analysis explains what statistical test has been used and why, additionaly authors prerformed PERMANOVA, Ktuskal-wallis rank sum test and post hoc Wilcoxon rank sum test . Authors also provided raw sequencing data -available in NCBI under the project number. The manuscript is written in standard english, no additional corrections are needed. Summarizng, the manuscript provides valiuable information on racoon dog's died by using advanced molecular tehniques, and thus more reliable than morphological and osteological expertise. Inventing RacBlock blocking oligonucleotide for 12S rRNA gene significantly increased the value of the paper.

RESPONSE: We appreciate the positive appraisal by the reviewer #1.

2. Once the Latin names have been given in the Results section there is no need to repeat them in the discussion, there are some repetitions e.g. line 266: Suidae (pigs or boars) is repeated in line 348, in this case the bracket should be dropped.

RESPONSE: We fixed them according to the advice.

Reviewer #2:

1. This paper investigates the diet of the raccoon dog using only a very limited number of samples, but the authors have successfully generated a blocking primer which is another achievement of this work. Perhaps the title could be modified so this is more explicit.

RESPONSE: We agree. The title was changed as follows: 

“Using DNA metabarcoding and a novel canid-specific blocking oligonucleotide to investigate the composition of animal diets of raccoon dogs (Nyctereutes procyonoides) inhabiting the waterside area in Korea”

2. Line 125- Please state clearly if these other canids coexist with raccoon dogs at sampled localities. This is mentioned in discussion but I think it is important for readers have this clarified earlier in the text

RESPONSE: To make it clearer that there is no possibility of the interference, we have removed ambiguous words such as "believe" and "thought" from the revised manuscript.

3. Line 144-146: this information is repeated and more complete in following sections

RESPONSE: The manuscript was revised to repeat the information in the following paragraphs. 

4. Line 150- How was the concentration of blocking primer optimized? PCR conditions are incomplete (extension?). Did the blocking primer anneal at the same temperature as the 12S universal primers? What are the sizes of the 12S and 16S libraries?

RESPONSE: We followed the thermal condition reported by Shehzad et al. (2012). The elongation step was not included in their study (Lines 152-153). 

According to the Tm Calculator of Thermo Fisher Scientific (https://www.thermofisher.com/kr/ko/home/brands/thermo-scientific/molecular-biology/molecular-biology-learning-center/molecular-biology-resource-library/thermo-scientific-web-tools/tm-calculator.html), Tm of 12SV5F and 12SV5R are 51.3℃ and 49.3℃, respectively. Tm of our blocking primer is calculated to be 63.7℃. The difference in Tm seems large. However, we believe it is not unusual according to previous research reports. For instance, using the same primer pair 12SV5F/12SV5R, Shehzad et al. (2012) and Kumari et al. (2019) successfully designed blocking oligonucleotides PrioB and OBS1, respectively, whose Tm are 60.9℃ and 65.4℃, respectively. We can see that their temperatures are similar to ours. In general, blocking oligonucleotides are designed to anneal before primers anneal. Therefore, it is reasonable that the annealing temperature of the blocking oligonucleotide is higher than the annealing temperatures of the primers.

The average size of targeted regions (excluding primer and adapter regions) of vertebrate 12S rRNA gene and invertebrate 16S rRNA gene are about 100 and 36 bp, respectively. The size of amplicons are larger since they also contain primer and adapter sequences (i.e., around 200 bp for 12S rRNA gene). 

Confirmation of inhibition of PCR amplification by the blocking oligonucleotide RacBlk was performed using a DNA extract from a carcass specimen of raccoon dog. The tested concentrations of RacBlk were 2, 3, 4, and 5 µM (see S2 Fig). We found that all concentrations of RacBlk tested successfully inhibited PCR amplification of raccoon dog DNA. To maximize the inhibitory efficiency, we chose 5 µM as the concentration of RacBlk. 

5. Line 156- again, PCR conditions look incomplete.

RESPONSE: We followed the thermal condition reported by De Barba et al. (2014). The elongation step was not included (Line 161). 

6. Line 168- Please state whether PhiX was used in the run

RESPONSE: Yes, 30% PhiX was also added (Line 169).

7. Line 176- I am not that familiar with this software but it seems minimum size of sequences considered was 80 bp for 12 S and 20 bp for 16S. I can imagine taxonomic results could be meaningful with 80 bp for 12S, but 20 bp for 16S looks impossible to me. I am also concerned about overestimation of OTUs as perhaps every unique 20bp sequence was considered as an OTU even if these 20 bp were included in larger sequences? How does the obiuniq command deal with size?

RESPONSE: We agree that short sequence reads can result in ambiguous identification, and we expect that the relatively high percentage of ambiguous reads for invertebrate 16S at the family level shown in Fig 2C is due to the short barcode region (36 bp, excluding primer and adapter regions). Meanwhile, we believe that the selection of 20 bp as a cut-off length is reasonable since the barcode region that was analyzed (excluding primer and adapter regions) is 36 bp (i.e., 16 bp as buffer length). The method to identify invertebrates by targeting the short barcode region of 16S rRNA gene (36 bp) was established by De Barba et al. (2014), and we followed their method. Using their method, we successfully identified invertebrates that are known to inhabit our sampling area (Fig. 2D). There may be an issue of overestimation of OTUs; however, this study limited the analysis to taxonomic identification only, and we did not perform species richness analysis. 

8. Line 183- I understand these data have some years now, but I think the authors should have used the currently available data at EMBL

RESPONSE: As suggested by the reviewers, we created the latest versions of the databases and reanalyzed. All results in the revised manuscript are the reanalysis results based on the latest versions of the databases. 

“Specifically, the taxonomic annotation was performed against 12S (vertebrate) and 16S (invertebrate) rRNA genes databases generated from the latest snapshot (updated on March 13, 2022) of EMBL nucleotide sequences (http://ftp.ebi.ac.uk/pub/databases/ena/sequence/snapshot_latest/std/).” Lines 184-187

In addition, the discussion on genera detected with new database was added (Lines 350-351, 358-359, 363-364, and 477-479), and the discussion on genera that were not detected with new database was removed.

9. Line 189- I think the comparison of libraries constructed with or without blocking primer should use data at OTU level. Otherwise important taxa or taxa of interest in future studies could be excluded and no results regarding this are reported (not available in supplementary materials). The fact that taxonomic assignment was achieved for most OUT at family level does not preclude analysis at genus and OTU level. Without this the Discussion section could be misleading as the authors clearly state: “indicating the absence of unintended inhibition of the amplification of preys’ DNA. “. I don’t understand how this was tested if analysis were performed at family level. Also why did you rarify these data? Was it to eliminate rare prey? I think metabarcoding data applied to diets, even using universal primers, implies data needs to be treated as presence/absence rather than abundance. There is a lot that cannot be controlled in a PCR reaction, namely different levels of primer affinity to each prey DNA, different digestibility of prey which can also bias DNA “quantity” etc. 

RESPONSE: We agree with the reviewer that OTU-level (rather than family-level) analysis is more relevant. We re-analyzed the results based on the number and composition of unique sequences (equivalent of OTUs) (Fig. 1). We confirmed that our conclusion was unchanged. There was no significant omission of unique sequences by RacBlk.

Rarefaction was performed only for the purpose to compare the sequencing results with and without RacBlk. No rarefaction was performed in other analyses, which was clarified in our revised manuscript (Lines 204-212). Meanwhile, rarefaction was necessary to compare the results with and without RacBlk. It was necessary because the number of sequences from prey animals (excluding sequences from raccoon dogs) was much smaller in libraries prepared without RacBlk than those with RacBlk. Since the difference in the number of sequences affects the comparison between the libraries, it was necessary to make the number of sequences the same across libraries by rarefaction. We rarefied all libraries to 120 reads, which was the smallest number of sequence reads of prey animals (excluding reads of raccoon dogs) found in the library prepared without RacBlk. Again, rarefaction was necessary since the comparison had to be done under the same condition (i.e., the same sequence depth or number of sequences).

We are aware of the argument of how to interpret sequence results. This issue was explicitly stated in the revised manuscript (Lines 466-471). Meanwhile, we wish to keep reporting sequencing results based on relative abundance as they also contain the information of detected or undetected. It is not possible to convert binary data to continuous data, but it is possible to convert continuous data to binary data. The information of relative abundance may provide some insights such as detection specificity and sensitivity (or insensitivity) by the sequencing method used. We would like to delegate to the readers how to interpret the results. Again, the issue of how to report sequence results (binary vs. continuous) was explicitly mentioned in the revised manuscript, however (Lines 466-471).

“In particular, we would like to acknowledge that there is ongoing argument over whether DNA-based methods can accurately quantify the proportion of dietary content. This study reported relative abundance rather than detected or undetected since the information of relative abundance also includes the information of detected or undetected and may provide some insights such as detection specificity and sensitivity (or insensitivity) of the sequencing method used.” Lines 466-471

10. Line 200- Differences between libraries were evaluated using BC distance. However this accounts for abundance and what I have mentioned before I don’t think this metric should be used as it likely results in spurious results. Unless the nº of OTUs from a particular family is used as an abundance metric? There are other distances that could be applied to presence absence data (Jaccard for example).

RESPONSE: We added the result based on Jaccard index, too. For the above reasons, we would also like to continue to include results based on Bray-Curtis dissimilarity, too. 

11. Line239- These are the kind of biases you would expect and that is why in my view, using abundance data in this kind of analysis is not correct

RESPONSE: As mentioned above, the issue of how to report sequence results (binary vs. continuous) was explicitly mentioned in the revised manuscript (Lines 466-471). 

12. Line 406- this sentence erroneously leads the reader to think these blocking primers only block raccoon dog DNA. I think this should be re-phrased

RESPONSE: The sentence was rephrased to tone down as follows:

“Furthermore, we could confirm that there was no significant change in dietary composition with and without RacBlk (Figs 1C and 1E), indicating that unintended inhibition of the amplification of preys’ DNA was insignificant.”

13. Line 413- I am very interested by this idea. Can you explain further why sequence length of blocking primer should affect the level of DNA blocked? If the primer was longer it may be more specific but I do not comprehend why there should be an implication with the amount of DNA blocked.

RESPONSE: The fact that longer blocking oligonucleotides are associated with higher blocking efficiencies has also been reported by previous studies (Boessenkool et al., 2012; Homma et al., 2022). However, we do not know the mechanism behind it. One possible explanation is that a short-length blocking oligonucleotide allows annealing and extension of the universal primer to the target region of the blocking oligonucleotide before the blocking oligonucleotide is annealed to it, resulting in amplification of the target organism (which should be blocked) by the universal primers. As a possible hypothesis, the following sentence was added.

“In fact, the fact that longer blocking oligonucleotides are associated with higher blocking efficiencies has been reported by previous studies [69, 70]. We do not know the mechanism behind, but one possible explanation is that a short-length blocking oligonucleotide allows annealing and extension of the universal primer to the target region of the blocking oligonucleotide before the blocking oligonucleotide is annealed to it, resulting in amplification of the target organism (which should be blocked) by the universal primers.” Lines 445-450

References

Boessenkool, S., et al. (2012). Blocking human contaminant DNA during PCR allows amplification of rare mammal species from sedimentary ancient DNA. Mol. Ecol., 21(8), 1806–1815.

De Barba, M., et al. (2014). DNA metabarcoding multiplexing and validation of data accuracy for diet assessment: application to omnivorous diet. Mol. Ecol. Resour., 14(2), 306–323.

Homma, C., et al. (2022). Effectiveness of blocking primers and a peptide nucleic acid (PNA) clamp for 18S metabarcoding dietary analysis of herbivorous fish. PLOS ONE, 17(4), e0266268.

Kumari, P., et al. (2019). DNA metabarcoding-based diet survey for the Eurasian otter (Lutra lutra): Development of a Eurasian otter-specific blocking oligonucleotide for 12S rRNA gene sequencing for vertebrates. PLOS ONE, 14(12), e0226253.

Shehzad, W., et al. (2012). Carnivore diet analysis based on next-generation sequencing: Application to the leopard cat (Prionailurus bengalensis) in Pakistan. Mol. Ecol., 21(8), 1951–1965.

---

## [Editor Report · Decision Letter 1]

24 Jun 2022

Using DNA metabarcoding and a novel canid-specific blocking oligonucleotide to investigate the composition of animal diets of raccoon dogs (Nyctereutes procyonoides) inhabiting the waterside area in Korea

PONE-D-22-06464R1

Dear Dr. Yamamoto,

We’re pleased to inform you that your manuscript has been judged scientifically suitable for publication and will be formally accepted for publication once it meets all outstanding technical requirements.

Kind regards,

Bi-Song Yue, Ph.D

Academic Editor

PLOS ONE

---

## [Editor Report · Acceptance letter]

28 Jun 2022

PONE-D-22-06464R1 

Using DNA metabarcoding and a novel canid-specific blocking oligonucleotide to investigate the composition of animal diets of raccoon dogs (*Nyctereutes procyonoides*) inhabiting the waterside area in Korea 

Dear Dr. Yamamoto:

I'm pleased to inform you that your manuscript has been deemed suitable for publication in PLOS ONE. Congratulations! Your manuscript is now with our production department. 

Kind regards, 

on behalf of

Dr. Bi-Song Yue 

Academic Editor

PLOS ONE